

# Lightweight-CancerNet: a deep learning approach for brain tumor detection

Asif Raza and Muhammad Javed Iqbal

Department of Computer Science, University of Engineering and Technology Taxila, Taxila, Punjab, Pakistan

## ABSTRACT

Detecting brain tumors in medical imaging is challenging, requiring precise and rapid diagnosis. Deep learning techniques have shown encouraging results in this field. However, current models require significant computer resources and are computationally demanding. To overcome these constraints, we suggested a new deep learning architecture named Lightweight-CancerNet, designed to detect brain tumors efficiently and accurately. The proposed framework utilizes MobileNet architecture as the backbone and NanoDet as the primary detection component, resulting in a notable mean average precision (mAP) of 93.8% and an accuracy of 98%. In addition, we implemented enhancements to minimize computing time without compromising accuracy, rendering our model appropriate for real-time object detection applications. The framework's ability to detect brain tumors with different image distortions has been demonstrated through extensive tests combining two magnetic resonance imaging (MRI) datasets. This research has shown that our framework is both resilient and reliable. The proposed model can improve patient outcomes and facilitate decision-making in brain surgery while contributing to the development of deep learning in medical imaging.

## INTRODUCTION

A tumor, characterized by uncontrolled cell division and the formation of an abnormal mass, can harm the brain, which consists of billions of cells. Cancer can develop either within or outside of the brain. This disorder exhibits the highest rates of morbidity and mortality compared to all other forms of cancer that affect both adults and children. Diagnosing brain tumors is a complex task due to the elusive nature of their beginnings and the challenge of accurately determining their growth rates (*Kleihues et al., 2002*).

Brain tumors are a significant health issue, with their occurrence rates consistently increasing. The National Cancer Institute (NCI) confirms that brain tumors are most commonly diagnosed in adults aged 50 or older (*National Cancer Institute, 2024*). By gender, they are found more predominantly in males (*Popat & Patel, 2022*). Unfortunately, brain tumors have become the leading cause of death in individuals below the age of 20 (*DeAngelis, 2001*).

In medical literature, enhancing the accuracy and efficiency of brain tumor detection has become a crucial factor. Intelligent diagnostic methods and treatment strategies are

Corresponding author
Asif Raza, asifraza.uet@gmail.com

essential for addressing this issue. The utilization of deep learning techniques for detecting brain tumors has received considerable interest, especially in the application of edge computing for medical prioritization systems. Deep neural networks, a key component of deep learning, have garnered the attention of researchers, engineers, and healthcare professionals. For instance, convolutional neural networks (CNNs) are commonly employed in medical image analysis, yielding promising results (*Gupta, Gaurav & Arya, 2024*; *Qureshi et al., 2022a*; *Cen et al., 2019*; *Gao et al., 2019*).

Magnetic resonance imaging (MRI) stands out as the go-to non-invasive technique encouraged by radiologists for scanning (*Ardan & Indraswari, 2024*; *Kale & Gadicha, 2024*; *Nawaz et al., 2021*; *Asiri et al., 2024*). It is beneficial for detecting subtle structural changes that might be difficult to detect with computed tomography (CT) scans (*Simo et al., 2024*; *Aamir et al., 2022*). However, identifying the specific type of tumor can be challenging, particularly when time is limited, as is often the case during the prediction stage of artificial intelligence-based solutions.

Real-time detection could expedite the diagnostic process compared to traditional methods (*Mahmud, Mamun & Abdelgawad, 2023*). Moreover, real-time detection during surgery could provide more precise information about the tumor's boundaries and location, allowing for accurate and potentially less invasive resections (*Mohan et al., 2022*).

While there have been significant advances in deep learning computing applications and services, we still face challenges in achieving real-time human-computer interaction with efficiency and the ability to detect unknown knowledge. In particular, clear advancements in ultra-light deep learning frameworks have yet to be observed in medical brain tumor diagnosis.

These challenges become even more critical during intraoperative brain surgery, where quick decisions are crucial. We have developed an ultra-light deep learning framework to address these gaps and provide an effective solution. This framework is intended to streamline tumor identification and enhance decision-making during brain surgery, ultimately improving patient outcomes.

The study of ultra-light deep learning frameworks (*Qureshi et al., 2022b*; *Chinaev et al., 2024*) in the medical brain tumor diagnosis presents two major challenges. Firstly, resource efficiency, *i.e.*, Ultra-light deep learning frameworks, must handle large medical imaging datasets efficiently, balancing real-time processing with limited computational resources. Secondly, maintaining diagnostic accuracy while reducing model complexity is challenging due to trade-offs between lightweight architectures and superior performance in brain tumor diagnosis.

Furthermore, focusing solely on the intelligent diagnosis of brain tumors overlooks the broader significance of how such technology can be integrated into medical triage platforms or used by doctors on mobile or advanced Reduced Instruction Set Computer (RISC) machine (ARM) devices.

Most of the research on brain tumor diagnostics has mainly concentrated on CNNs within deep learning networks (*Lee, Chae & Cho, 2024*; *Vinod, Prakash & Salman, 2024*; *Tinu, Appathurai & Muthukumaran, 2024*). A widely used deep learning network can be employed to extract features from numerous medical images. Nevertheless, there is a lack

of research examining the realistic implementation of lightweight models to diagnose brain tumors (*Chengwei, 2019*; *Howard et al., 2017*; *Qiang et al., 2021*). Most of the deep learning frameworks require substantial computational power and resources. Furthermore, deep learning algorithms fail to achieve a balance between accuracy and speed. This research presents a lightweight deep learning model called Lightweight-CancerNet as a solution to address the existing challenges. This model is designed to be utilized by ARM devices.

It is suggested that the customized Lightweight-CancerNet be used with MobileNet as the backbone, which will help extract the distinguished features from MRI images. The NanoDet is used as a head for the detection of cancerous parts. We have demonstrated the strength and reliability of our proposed model by conducting extensive experiments on tumor samples.

The notable contributions of our work are as follows:

- An accurate method capable of computing reliable image features to enhance the tumor's detection is proposed.
- A robust framework is designed to improve the mAP and accuracy to detect the cancer, tumor, or lesions.
- A comprehensive evaluation of the proposed framework is performed on a complex dataset, and its effectiveness is confirmed through rigorous experimentation.

The remainder of the article is divided into four sections. "Literature Review" covers the existing work; "Methodology" discusses the proposed methodology. In "Experimental Setup and Results" and "Conclusion", the results and the conclusion are presented.

## LITERATURE REVIEW

This section has examined previous studies on detecting and categorizing cancers from medical images. Additionally, we analyzed various object detection models, analyzing their performance, strengths, and weaknesses. We have evaluated each model based on mean average precision (mAP), precision, recall, and F1 score.

Brain tumor segmentation techniques based on machine learning typically rely on voxel-based features extracted from the image's volume of interest (VOI). Various segmentation methods have been evaluated, demonstrating diverse performances (*Kumar, 2023*). *Abdusalomov, Mukhiddinov & Whangbo (2023)* employed You Only Look Once (YOLO)v7 and transfer learning techniques to enhance brain tumor diagnosis in MRI scans, achieving an exceptional 99.5% accuracy in detecting prevalent brain tumor types like Glioma, Meningioma, and Pituitary. However, they recognize the need for further investigation, especially in determining minor tumors. The methods ResNet (*Aggarwal et al., 2023*) and CRF-RNN were employed to achieve precise segmentation outcomes.

The evaluation of performance was conducted utilizing the Multimodal Brain Tumor Segmentation Benchmark (BraTS) dataset (*BraTS, 2020*). Similarly, *Ghafourian et al. (2023)* integrated the results of SVM, Naive Bayes, and k-nearest neighbors (KNN). They combined the outcomes of multiple models by averaging their values, thereby enhancing tumor classification results.

The BraTS dataset has a significant impact on brain tumor imaging studies. *Menze et al. (2015)* introduced BraTS as a standardized dataset with MRI scans and tumor annotations. *Bakas et al. (2017)* augmented BraTS by incorporating expert annotations and radiometric characteristics. Further, *Bakas et al. (2019)* identified machine-learning algorithms for tumor segmentation and prediction.

Furthermore, segmentation and diagnosis studies using MR images were conducted. This research employed a search algorithm to apply a thresholding technique. Morphological operations and connected component analysis were employed to minimize image noise and enhance brain tumor identification. Comparisons with CNN algorithms revealed high success rates in the obtained results (*Aleid et al., 2023*).

The concept of lightweight models has emerged in recent years, aiming to address model size and speed issues. Unlike working with pre-trained models, designing lightweight models offers an alternative approach. *Shelatkar et al. (2022)* introduced a novel way of diagnosing brain tumors using a lightweight deep-learning model with a fine-tuning methodology. The dataset acquired from the RSNA-MICCAI brain tumor radiogenomic classification was utilized in this investigation. The preprocessed data is partitioned into several subsets for testing and training the model. The accuracy of the YOLOv5 model is reported to be 88% (*Shelatkar et al., 2022*).

MobileNetV1 introduces a significant advancement in depth-wise separable convolution. The network forgoes conventional standard convolutions. The tumor detection accuracy of MobileNet has been reported to be 82.61% (*Ullah et al., 2022*).

Single Shot Detector (SSD) is a scanner that detects a single event. The proposed network does not include a region proposal; instead, it predicts the bounding boxes and the classes directly from feature maps in a single run. SSD incorporates small convolutional filters to enhance precision and predict object classes and offsets to the default bounding boxes (*Cen et al., 2019*). The SSD has an accuracy rate of 82.7% in detecting cancer on endoscopic images.

*Ronneberger, Fischer & Brox (2015)* proposed the U-Net architecture, which has now become a standard for medical image segmentation tasks, specifically in brain tumor detection. The U-Net model accomplished a precision rate of 93.4% when evaluated on the BraTS 2013 dataset, as reported by *Menze et al. (2015)* in 2015. However, the system's computational complexity and memory constraints hinder its implementation on devices with limited resources.

To overcome this constraint, *Howard et al. (2017)* introduced MobileNet, a CNN design that is less resource-intensive yet achieves similar accuracy to U-Net. MobileNet employs depthwise separable convolutions to decrease computational complexity and memory consumption, resulting in a 91.2% accuracy rate on the BraTS 2017 dataset (*Howard et al., 2017*).

In 2018, *Zhang et al. (2017)* presented ShuffleNet, a lightweight CNN architecture with enhanced efficiency and accuracy. ShuffleNet employs group convolutions and channel shuffle operations to decrease computational complexity and memory consumption, resulting in an accuracy of 92.5% on the BraTS 2013 dataset.

In 2017, *Iandola et al. (2016)* introduced SqueezeNet, a very efficient CNN design that offers cutting-edge performance while demanding minimal computational resources. The SqueezeNet utilizes knowledge distillation to train a compact model that emulates the functionality of a pre-existing, more intricate model, resulting in 94.1% accuracy on the BraTS 2017 data.

Additional research has investigated the application of deep learning models for the identification of brain tumors, such as DeepMedic (*Kamnitsas et al., 2016*), 3D U-Net (*Çiçek et al., 2016*), U-Net++ (*Zhou et al., 2018*), and VGG-16 (*Simonyan & Zisserman, 2015*). These models have attained different levels of precision. However, it frequently requires substantial computer resources and memory.

Despite achieving a reasonable level of accuracy, there remains a possibility for further enhancement in detecting brain tumors. Although techniques have potential, their ability to analyze various datasets and clinical settings may need to be revised. Furthermore, further improvements are required to enhance accuracy and ensure reliable diagnosis in real-world situations.

## METHODOLOGY

Efficient and effective automated localization and detection of cancer from medical images, *i.e.*, CT, endoscopy, MRI X-rays, *etc.*, is still a complex task because of the vast slight variations in the size, color, and position of lesions. Moreover, medical investigations require lightweight models that can be used through mobile or arm devices. In this work, we tried to overcome the challenges above by fine-tuning an efficient model. The proposed method is robust and lightweight, providing better mAP detection for cancer and lesions in medical images.

The presented work is composed of two main components. The first is 'Data Preparation', and the second is a deep learning-based model for cancer detection. The first module assembles two datasets to locate the exact region of interest, while the second module of the new model presents a cancer/lesion detection model. This module consists of two sub-modules: the first is the study of different lightweight models used for cancer detection, and the second will be the training component, which performs training using the critical points computed from the model. Figure 1 illustrates the flow diagram of the proposed methodology, and Fig. 2 presents the overview of the proposed work.

The steps for a lightweight cancer detection model are depicted in the flow diagram. It starts by taking an input image dataset, cleaning and preparing it, and then extracts key feature vectors using a MobileNet backbone. These features are further processed by a path aggregation network (PAN) before reaching the NanoDet detection head, which identifies and locates objects in the image. Finally, the trained model is used to analyze detection results for inference.

### Datasets

For training the model, we have used two different publicly available datasets. One is the Multimodal Brain Tumor Segmentation Benchmark (BraTS) (*Bakas, 2018*) with 1,140 images. It is a publicly available brain tumor dataset that contains four MRI modalities,

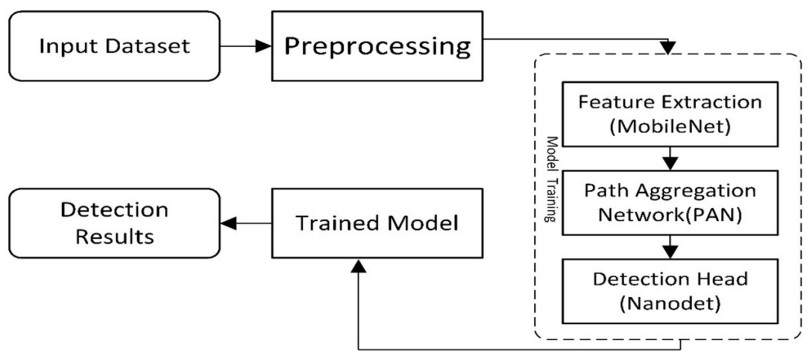

**Figure 1 A visual overview of the Lightweight-CancerNet framework's methodological flow, illustrating each stage from data preprocessing to model training and evaluation.**

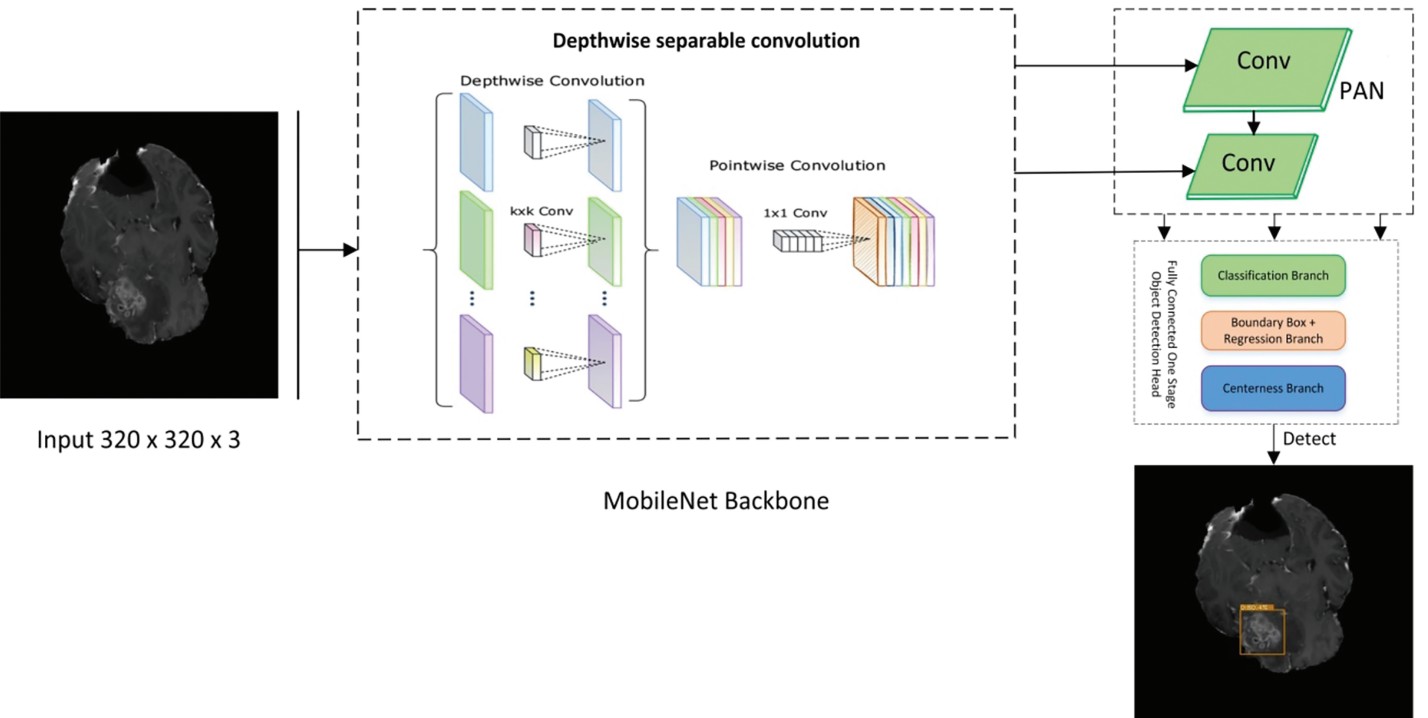

**Figure 2 Proposed lightweight cancernet with three modules feature extraction by using MobileNet backbone, PAN for the fusion of feature maps, and NanoDet head as prediction module.**

having T1, T1ce, T2, and Flair. BraTS is used extensively by researchers (*Ardan & Indraswari, 2024*; *Bakas et al., 2019*; *Kumar, 2023*). The other one is RSNA-MICCAI competition data (*Roberts, 2021*) having 400 images. The dataset is also publically available and has images in various resolutions, colors and modalities. The dataset has been expanded 1,131 images using horizontal and vertical flip augmentation. We ensured that the augmented images would not be part of test data so that they could not effect the test

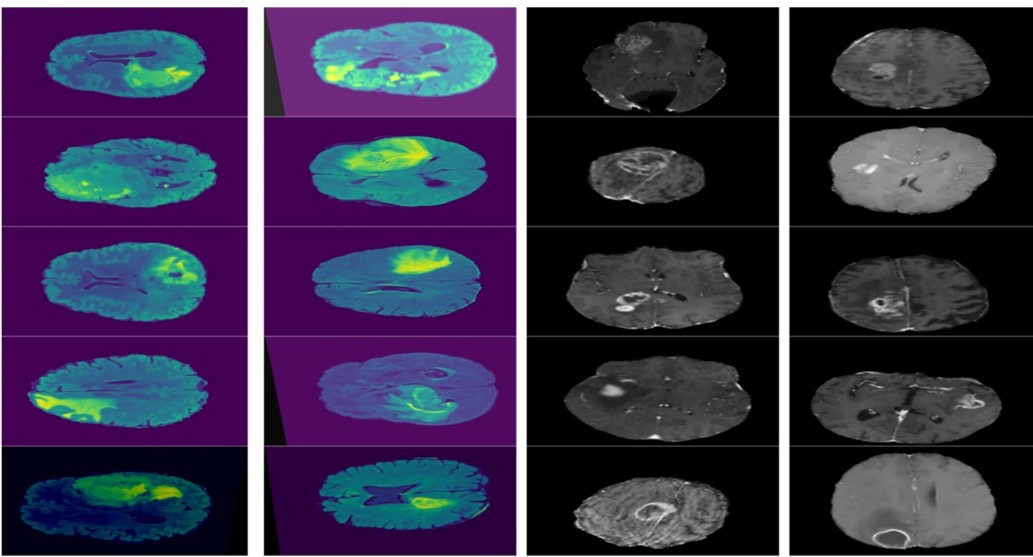

**Figure 3** Displays example MRI slices with cancerous regions, offering insight into the dataset and visual characteristics used for training and testing the model.

results of the model. Both datasets were merged to enhance the accuracy and robustness of the findings.

Furthermore, combining datasets increases the diversity and sample size, leading to reliable conclusions. The images in the collection have a resolution of 320 × 320. Figure 3 shows the sample images from the dataset.

## Annotation

In this work, first, we used the dataset's images as input images and annotated the regions of interest using the coordinates given with the datasets. Annotation is done manually using the annotation tool available at https://www.makesense.ai/. In the annotation process, positive labels and bounding box coordinates were indicated. After annotation, the images were forwarded to the MobileNet backbone for feature extraction.

## MobileNet

The MobileNet architecture is commonly employed as the basis for object detection algorithms because of its exceptional efficiency and accuracy. Therefore, it is used as the backbone of the proposed framework. The backbone network of MobileNet is composed of fully connected (FC) layers followed by many depth-wise separable convolution layers. According to Eq. (1), the MobileNet backbone network can accept an input image, W × H × D as input size.

$$Imgsize = W \times H \times D \tag{1}$$

The MobileNet backbone network produces a feature map with dimensions H × W × D, where W is the width, H is height, and D is the number of feature channels. The Region Proposal Network (RPN) and object detector utilize this feature map as input to recognize

and accurately identify items in the picture. The RPN produces a sequence of region suggestions. Then, the feature map obtained from the MobileNet backbone network is passed *via* Region of Interest (ROI) pooling to extract a feature vector of a consistent size for each suggestion. The ROI pooling method involves applying a max-pooling operation to each rectangular region of the feature map that corresponds to the region suggestions.

The rationale behind the selection of MobileNet as the backbone is the selection of MobileNet as the backbone network is a well-reasoned choice that balances the need for computational efficiency, model performance, and adaptability to various mobile and embedded vision applications (*Howard et al., 2017*).

### Path aggregation network (PAN)

PAN is used to fuse different features received from different paths from the MobileNet backbone. The fusion is done using the concept of element-wise addition. Equation (2) shows the simplified working of PAN.

$$Ffused = Fi + Fi^{th} \in \mathrm{R}H \times W \times C \tag{2}$$

$Fi \in \mathrm{R}\ H \times W \times C$ are the feature maps from the $i^{th}$ path, where $H, W, C$ are the height, width, and number of channels, respectively. The final output of the PAN module is the fused feature map $F_{fused}$, which is then passed to the next network stage.

Then, the fine-tuning of hyperparameters like learning rate, hidden layers, neurons, and batch sizes is carried out. After that, we trained the NanoDet model and tested the final lightweight model against the hold-out dataset.

### NanoDet

NanoDet is an advanced object detection model that does not require anchors and has various remarkable features. As an anchor-free model in the Fully Convolutional One-Stage Object Detection (FCOS) technique (*RangiLyu, 2021*). FCOS consists of three main branches. The three branches of FCOS are:

**Classification Branch:** It consists of a series of convolutional layers responsible for predicting the class object at each location on the feature map. The output layer employs softmax or sigmoid function to predict the class probabilities. Furthermore, it does not only predict a single class but also provides probability scores for the classes that allow it to handle multiclass problems.

**Bounding Box Branch:** It is responsible for the offset prediction for the bounding box coordinates relative to each location of the feature map. It predicts the distances to the sides of the bounding box that encapsulates the detected object. These distances are later transformed into absolute bounding box coordinates in the original image space.

**Center-Ness Branch:** This cognitive function differentiates the position of an object and its central point. The model aids in reducing the impact of low-quality bounding boxes that are distant from the object's center by calculating a centeredness score for each feature level. Greater normalized values indicate closer proximity to the center of the object. The center-ness score is high if the location is near the object's centre and low if it is near the edges.

In addition, NanoDet is exceptionally lightweight, with a model file size of either 980 KB (INT8) or 1.8 MB (FP16). Its lightning-fast speed of 97 fps (10.23 ms) on mobile ARM CPU, in addition to this, makes it an excellent option for real-time object identification apps (*RangiLyu, 2021*). The current article features a straightforward network layout and reduced network parameters, facilitating its portability and deployment.

NanoDet is used as a head, and convolutional layers for classification and regression were used. Each convolution layer comprises a Convolutional Layer, Norm Layer, and Activation Layer. These are applied to the output of the backbone, *i.e.*, input feature maps in parallel. Then, the output is generated using Eqs. (3) and (4).

$$cls\_feat = \sigma((Wcls * x) + bcls) \tag{3}$$
$$reg\_feat = \sigma((Wreg * x) + breg) \tag{4}$$

where x is the input feature map, *Wcls* and *Wreg* are the weights of the convolutional layers for classification and regression, respectively. bcls and breg are the convolutional layers biases for classification and regression. Here, $\sigma$ is the activation function (LeakyReLU) and * denotes the convolution operation.

The modified NanoDet Head maintains the fundamental architecture of the initial model, consisting of a Convolutional Module (ConvModule) followed by two distinct convolutional layers (conv_cls and conv_reg) for classification and regression tasks, respectively. However, we present the subsequent enhancements:

1) **Reduced number of channels:** We have decreased the number of channels in the ConvModule from 256 to 128, reducing computational costs while maintaining essential features.

2) **Depthwise separable convolutions:** We replace the standard convolutional layers with depthwise separable convolutions, which factorize standard convolutions into depthwise and pointwise convolutions, reducing the number of parameters and computations.

The optimized convolutional layer can be represented as:

$$Y = Conv(F, K, P, C, G) = \delta(\Sigma(Gi * F)) \tag{5}$$

Let F be the input feature map, K be the kernel size, P be the padding, C be the number of channels, and G be the number of groups. Y is the output feature map, Gi is the group convolutional kernel, $\delta$ is the activation function (ReLU), and $\Sigma$ denotes the summation over the groups.

The model significantly reduces computational time by introducing these optimizations, making NanoDet more suitable for real-time object detection applications while maintaining accuracy.

The pseudo-code of the proposed Lightweight-CancerNet algorithm is listed in Algorithm 1.

| Algorithm 1 Lightweight-CancerNet (Proposed). |
| :--- |

**Input:**

   1. D $= \{(\mathbf{X}i, \mathbf{y}i)\}_{i=1}^{N}$, where $\mathbf{X}i \in R^{w \times h \times 3}$ is the i^th input image and $\mathbf{y}i \in \{0, 1\}$ is the corresponding label

   *A*, **Steps:**

   2. **Feature Extraction**

       • $F \leftarrow MobileNet(D)$, Extract features from the annotated images in *D* using MobileNet to form *F*

   3. **Path Aggregation Network (PAN)**

       • $F' \leftarrow PAN(F)$, Apply the Path Aggregation Network (PAN) to the features *F* to form *F'*

   4. **NanoDet Model**

       • $M \leftarrow NanoDet(F', optimized = True)$, Train the NanoDet model with the optimized parameter settings using the features *F'* and the annotated dataset $D^*$ to form the trained model *M*

   5. **Training**

       • $M \leftarrow Train(M, D^*, hyperparameters)$, Train the model *M* with the hyper parameters and the annotated dataset $D^*$

**Output:**

**M**, a trained Lightweight-CancerNet model

## EXPERIMENTAL SETUP AND RESULTS

The experiments utilize the Pytorch deep learning framework, built in Python. The algorithm training environment is a Windows 10 operating system, running on an Intel(R) Core (TM) i5-1235U@ 2.64 GHz and an NVIDIA UHD Graphics 630 Ti 08 GB. Based on the 7:2:1 ratio, the dataset is randomly partitioned into the training, test, and validation sets. The training begins with an initial learning rate of 0.11 and is set at 0.14 after some attempts. Before training, warm-up training is conducted with a step size of 5. Simultaneously, one-dimensional linear interpolation (*Hung & Hung, 2014*) is employed. The cosine annealing algorithm (*Zhang et al., 2023*) adjusts the learning rate dynamically during the training process.

Moreover, a training step count of 300 and a batch size of 32 based on the specific computer configuration were employed for the training process. The model's input image size during training is 320 × 320. The detection model's robustness to object size is enhanced to a certain degree by training on photos of varying scales. It took 7 h for training. Table 1 exhibits the training parameters used in this study.

Figure 4 depicts the accuracy and learning curve of the model throughout 300 epochs. It exhibits an initial drop in the first 20 epochs. It is possibly due to initialization. It then shows an increase throughout the training process, which achieved an accuracy of 98%.

### Evaluation parameters

We employed the precision and recall rates as fundamental metrics regarding recognition accuracy. The precision rate assesses the correctness of the prediction, while the recall rate examines the completeness of the search. Equations (6) to (9) calculate the precision rate, recall rate, accuracy, and F1 score, respectively. The mAP is subsequently computed as the

**Table 1 Training parameters for Lightweight-CancerNet model.** Detailed parameter settings used during the training phase of the Lightweight-CancerNet model, including batch size, learning rate, and number of epochs, offering insight into the experimental setup for reproducibility.

| Framework parameter | Value |
|---|---|
| Epochs | 300 |
| Batch size | 32 |
| Learning rate | 0.14 |
| Image size | $320 \times 320$ |
| Optimization function | AdamW |
| Loss function | Huber loss (IOU loss) |

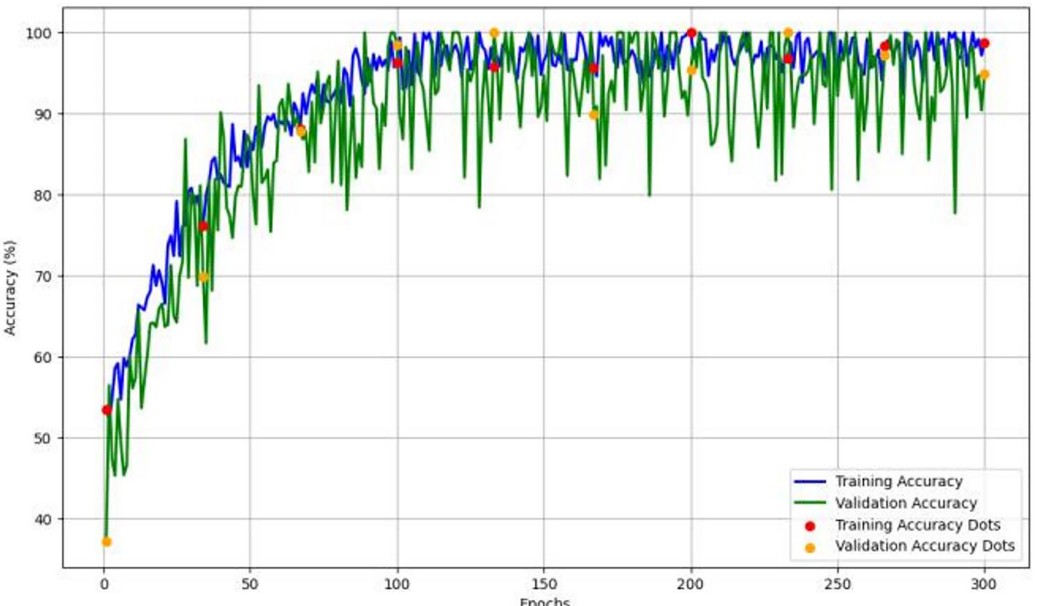

**Figure 4 The model's training and validation accuracy over 300 epochs, highlighting trends that indicate the model's learning stability and generalization capability across epochs.**

final evaluation index of accuracy, relying on the precision rate and recall rate. The mAP metric is employed to assess the overall performance of the trained model across all categories. The threshold for determining detection success is set at 0.5, determined by the intersection and union ratio.

$$Precision = \ TP/(TP + FP) \tag{6}$$
$$Recall = TP/(TP + FN) \tag{7}$$
$$Accuracy = \ (TP + TN)/(TP + TN + FP + FN) \tag{8}$$
$$F1 = (2 \ \times TP)/(2 \times TP + FN + FP) \tag{9}$$

In the formula, true positive (TP) denotes the total of predicted positive and actual positive samples, false negative (FN) denotes the total of predicted negative and actual

positive samples, true negative (TN) refers to the total number of instances that were correctly predicted as negative, and false positive (FP) denotes the total of projected positive and actual negative samples.

The equation for calculating the mAP is presented in Eq. (10), where AP represents the average accuracy of each group, t refers to the analyzed image, and T represents the total number of test photographs.

$$mAP := \sum_{i=0}^{T} AP(t)/T \tag{10}$$

## Detection results on BraTS dataset

The proposed framework is compared with other state-of-the-art models on the BarTS dataset. Table 2 shows that the proposed Lightweight-CancerNet model demonstrates higher performance than state-of-the-art models on the BraTS dataset, obtaining an accuracy of 97.3% on BraTS. Significantly, this surpasses the 93.4% accuracy attained by U-Net, the 91.2% accuracy of MobileNet, the 92.5% accuracy of ShuffleNet, and the 94.1% accuracy of SqueezeNet. The substantial enhancement in precision showcases the effectiveness of the proposed model on the BraTS dataset. Figure 5 illustrates the detection results of BraTS dataset.

## Comparison of the lightweight CancerNet with other state-of-the-art the art deep learning models

An accurate and nominative set of features is required for effective object detection. Therefore, we evaluated the results of Lightweight-Cancernet with other deep learning feature extractor frameworks including YoloV5 (*Shelatkar et al., 2022*), MovileNet (*Ullah et al., 2022*), SSD (*Cen et al., 2019*), U-Net (*Ronneberger, Fischer & Brox, 2015*), ShuffleNet (*Zhang et al., 2017*), SqueezNet (*Iandola et al., 2016*), DeepMedic (*Kamnitsas et al., 2016*), 3D-UNet (*Çiçek et al., 2016*) and VGG-16 (*Simonyan & Zisserman, 2015*). The results of the merged dataset are used for comparison to enhance the generalizability and robustness. The proposed model was inference by using the same conditions and hardware, *i.e.*, used for the training. Figure 6 represents cancer detection results using the proposed framework Lightweight-Cancernet framework. Moreover, Table 3 presents the analyses of various object detection models, including their respective accuracy, mAP ratings, and time taken to detect a single image.

YOLO earns an accuracy of 88% and takes 0.05 s, demonstrating its efficacy in detecting objects in real time. ShuffleNet, recognized for its effectiveness on mobile devices, attains a marginally superior accuracy of 92.5%. The MobileNet, with one instance achieving an accuracy of 82.61% and another earning a far higher score of 91.2%. This highlights how specific implementation details or dataset characteristics influence the performance. The U-Net demonstrated an accuracy of 93.4%, suggesting there may be compromises between accuracy and speed.

**Table 2 The proposed Lightweight-CancerNet framework's performance with other state-of-the-art models, highlighting accuracy, to illustrate its competitive efficacy.**

| Model | Dataset | Accuracy |
|---|---|---|
| U-Net | BraTS | 93.4% |
| MobileNet | BraTS | 91.2% |
| ShuffleNet | BraTS | 92.5% |
| SqueezeNet | BraTS | 94.1% |
| Proposed | BraTS | 97.7% |

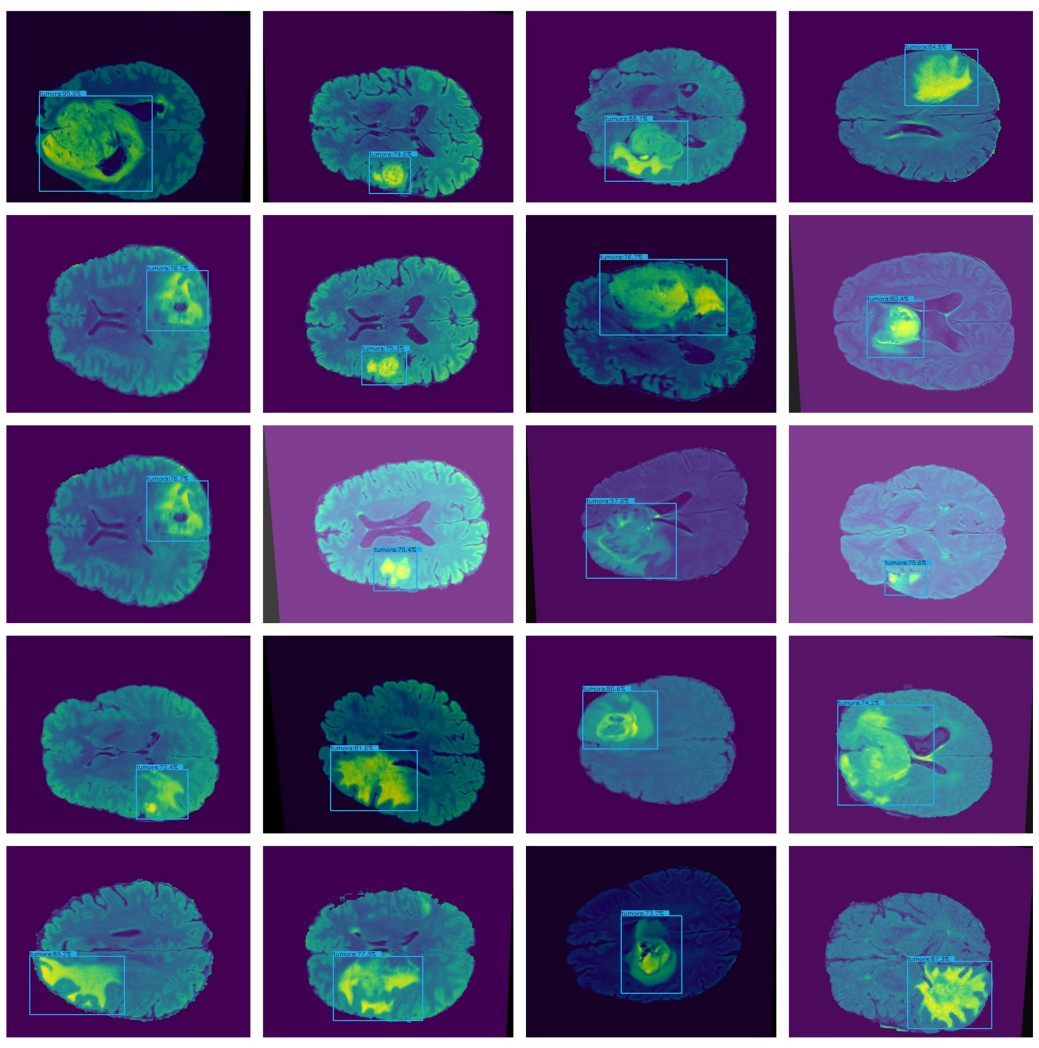

**Figure 5 Lightweight-CancerNet framework on BraTS.**

Nevertheless, the Lightweight-CancerNet stands out as the top performer in the chart, presenting an amazing mAP of 93.8%, indicating notable progress in precision and effectiveness, resolving the compromises observed in earlier models.

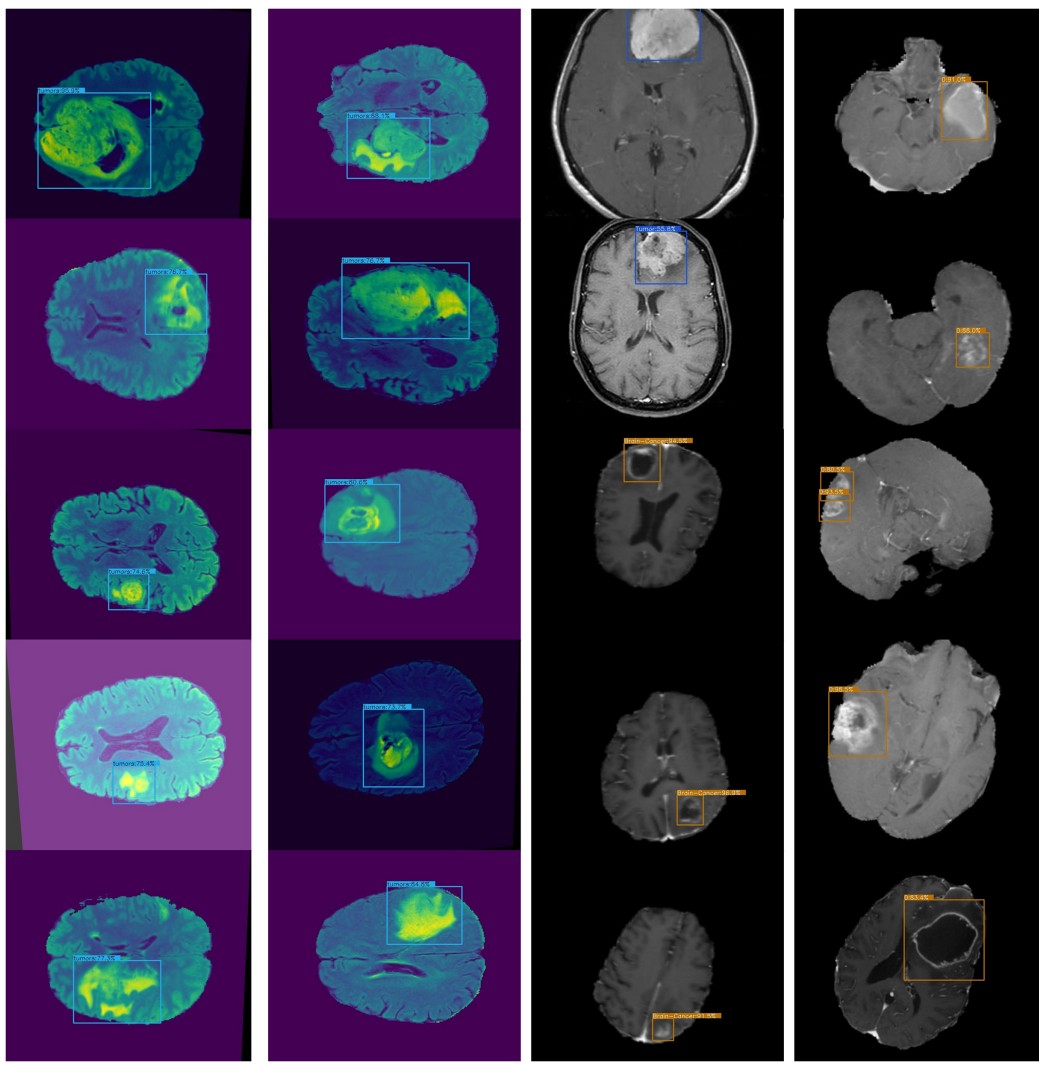

**Figure 6** Presents sample outputs showing successful detection of cancerous regions.

Moreover, the experimental results also demonstrate the exceptional effectiveness of the proposed Lightweight-CancerNet model for segmenting brain tumors. The suggested model surpasses previous state-of-the-art models, such as YoloV5, MobileNet, SSD, U-Net, ShuffleNet, SqueezNet, DeepMedic, 3D-UNet, and VGG-16, with an accuracy of 98% and an mAP of 93.8%. The suggested model attains remarkable accuracy and mAP while using considerably less computational effort, averaging around 0.003 s per image. This model is far more efficient than previous models, such as U-Net, which has a processing time of 3.33 s per image. Figure 7 represents a graphical view of performance evaluation concerning different threshold values.

## Lightweight-CancerNet performance: threshold analysis

This section presents a comprehensive evaluation of the performance metrics of Lightweight-CancerNet for brain tumor detection. The results illustrate the method's

**Table 3 Detailed performance comparison of Lightweight-CancerNet with alternative models.**
Expands on performance metrics across different state-of-the-art models, presenting accuracy, and computational efficiency to assess Lightweight-CancerNet's suitability for efficient and accurate cancer detection.

| Method | MAP | Accuracy | Inference time (S) per image |
|---|---|---|---|
| YoloV5 | – | 88% | 0.05 |
| MobileNet | – | 82.61% | 0.007 |
| SSD | 77.2% | 82.7% | 0.016 |
| U-Net | – | 93.4% | 3.33 |
| MobileNet | – | 91.2% | 0.01 |
| ShuffleNet | – | 92.5% | 0.0074 |
| SqueezNet | – | 94.1% | 0.0071 |
| DeepMedic | 74.4% | – | – |
| 3D-UNet | 75.1% | – | – |
| VGG-16 | 71.8% | – | – |
| Lightweight-CancerNet | 93.8% | 98% | 0.003 |

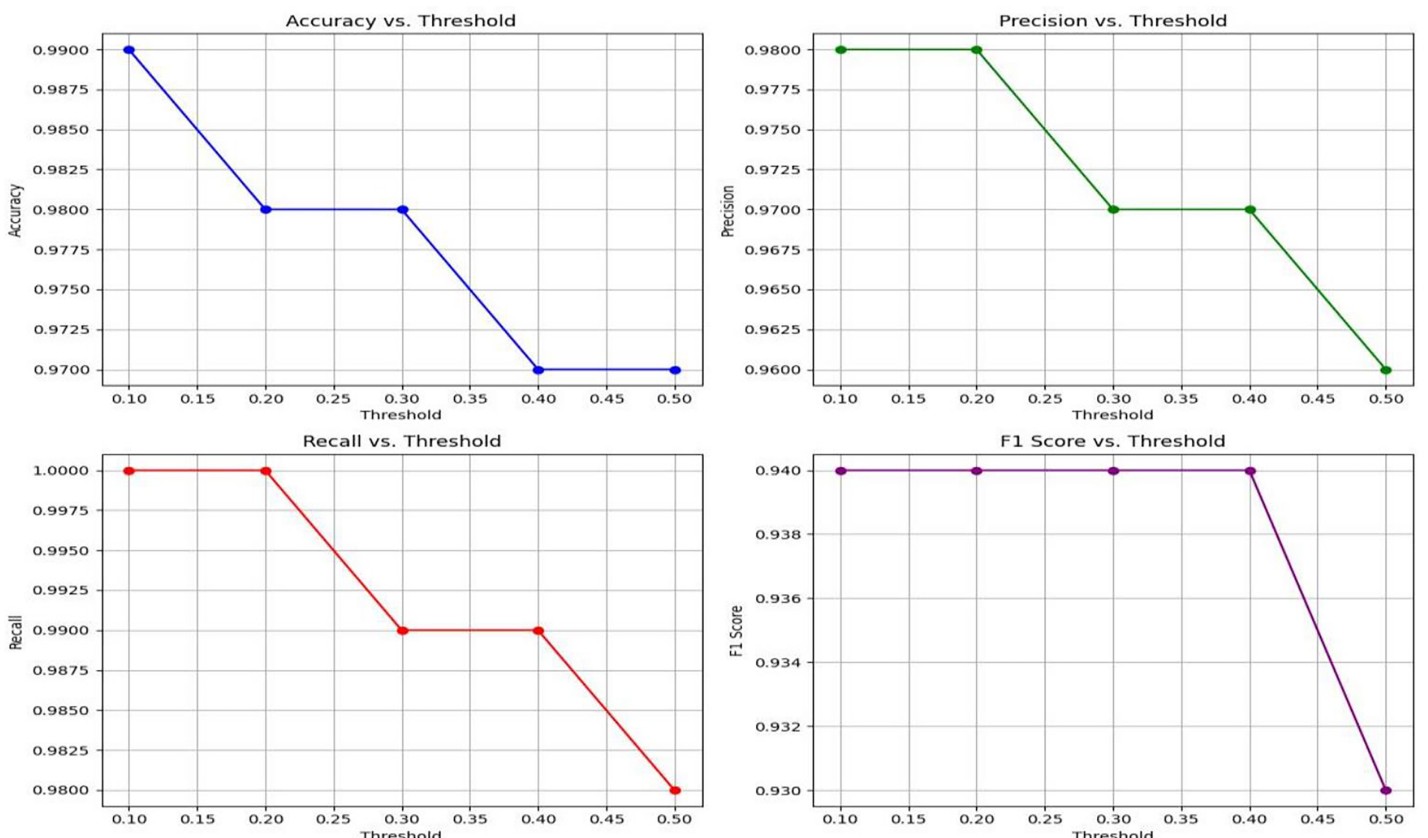

**Figure 7 A visual representation of how different classification thresholds affect performance metrics.**

**Table 4 Performance metrics for brain tumor segmentation at various thresholds.** Presents segmentation performance across multiple thresholds, showing variations in accuracy, recall, precision and F1 score to assess the model's robustness in different scenarios.

| Threshold | Accuracy | Recall | Precision | F1 score |
|-----------|----------|--------|-----------|----------|
| **0.1** | 0.99 | 1.00 | 0.98 | 0.94 |
| **0.2** | 0.98 | 1.00 | 0.98 | 0.94 |
| **0.3** | 0.97 | 0.99 | 0.97 | 0.94 |
| **0.4** | 0.97 | 0.99 | 0.97 | 0.94 |
| **0.5** | 0.96 | 0.98 | 0.96 | 0.93 |

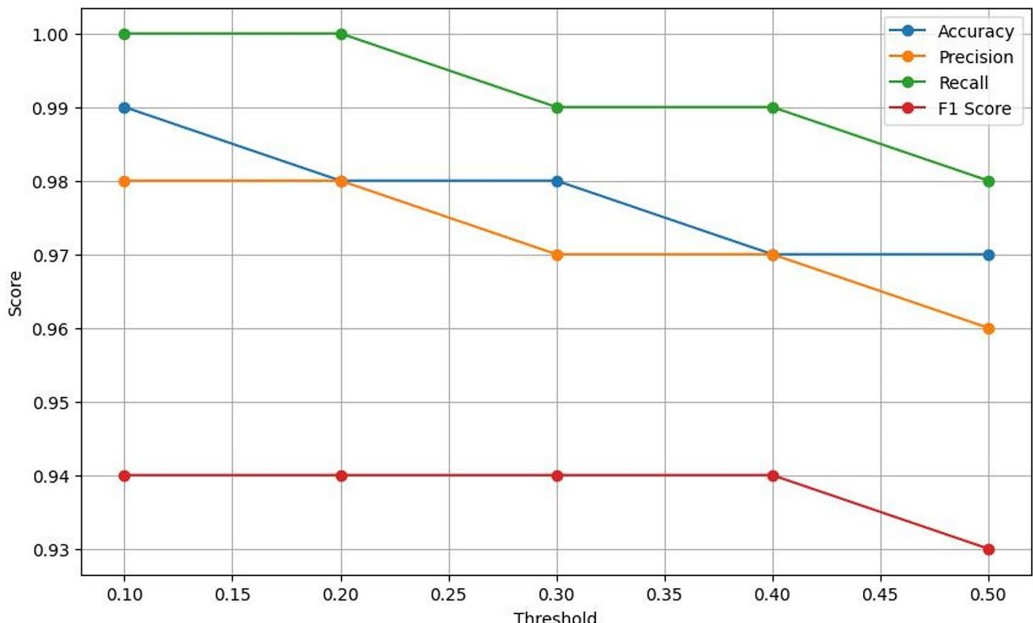

**Figure 8 Plots the four key performance metrics against threshold values, helping visualize trade-offs between metrics and guiding threshold selection for the most balanced performance.**

reliability in detecting brain tumors, considering the balance between accuracy and the choice of threshold (*Kumar, 2018*). Here the threshold is the value or boundary that used to determine that the input will be categorized in positive or negative with respect to defined value.

Table 4 exhibits that Lightweight-CancerNet attains a high level of accuracy (between 97% and 99%) consistently over a range of thresholds, proving that it is a reliable method for identifying brain cancers. Additionally, there is a trade-off between precision and threshold, beginning at a high of 98% for a 0.1 threshold and dropping to 96% for a 0.5 threshold. This indicates that CancerNet is better at detecting cancers at low levels, although it may miss some crucial instances due to this. Meanwhile, recall remains high against all threshold values, decreasing between 98% and 1.0. Therefore, CancerNet can

find all pertinent brain tumor instances. A plot with precision, recall, accuracy, and F1 score curves plotted against thresholds is depicted in Fig. 8.

The F1 score shows a slight drop as the threshold increases. Starting at a threshold of 0.1, it approaches 94%; at 0.5, it hits 93%. CancerNet has great potential in accuracy and recall, implying that it can accurately identify and capture many brain cancers. Setting the threshold correctly requires examining the context, as there is a trade-off between accuracy and threshold. If the aim is to minimize false positives, then reducing the threshold would be the most effective approach. However, opting for a more significant threshold may be the best option if the main objective is to detect all potential cancers.

## CONCLUSION

This article presents Lightweight-CancerNet, a new deep-learning framework that provides excellent accuracy and efficiency in the diagnosis of brain tumor. Our model surpasses existing models in terms of accuracy and efficiency, utilizing the capabilities of MobileNet and NanoDet. The proposed Lightweight-CancerNet is designed to manage extensive medical imaging datasets using minimal computer resources, making it highly suitable for instantaneous processing. This study has significant implications for improving patient outcomes and decision-making in brain surgery. It also illustrates the potential of using lightweight models to detect objects in medical imaging quickly.

The proposed model's performance heavily depends on the training dataset's diversity and quality. Biased or limited datasets may result in suboptimal performance. Another limitation of the model is that it may not generalize well to unseen data and require more parameter training.

In our future research, we will explore more improvements and extensions to our framework and apply it to other medical imaging applications. Furthermore, we intend to evaluate the application of our system in real-world clinical settings by examining its effectiveness in practical scenarios and its potential to improve patient outcomes.

## ACKNOWLEDGEMENTS

ChatGPT was used to edit the "Annotations" section and to understand the reviewer questions.

### Funding
The authors received no funding for this work.

### Competing Interests
The authors declare that they have no competing interests.

### Author Contributions
- Asif Raza conceived and designed the experiments, performed the experiments, analyzed the data, performed the computation work, prepared figures and/or tables, authored or reviewed drafts of the article, and approved the final draft.

- Muhammad Javed Iqbal conceived and designed the experiments, analyzed the data, authored or reviewed drafts of the article, and approved the final draft.

## Data Availability

The BraTS (Brain Tumor Segmentation) dataset is a widely used benchmark in the field of medical image analysis, particularly for brain tumor detection and segmentation tasks. It was introduced by the Medical Image Computing and Computer-Assisted Intervention (MICCAI) community and is updated regularly to reflect advancements in brain imaging and tumor segmentation techniques. It is available at https://www.med.upenn.edu/cbica/brats2020/data.html. The BraTS dataset is freely accessible for research purposes through the MICCAI official site, with a license for academic use only.

Access to the BraTS dataset can be requested here https://www.med.upenn.edu/cbica/brats2020/registration.html and questions directed to brats2020@cbica.upenn.edu

The code is available in the Supplemental Files.

## Supplemental Information

Supplemental information for this article can be found online at http://dx.doi.org/10.7717/peerj-cs.2670#supplemental-information.

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
