# Peer review of "Lightweight-CancerNet: a deep learning approach for brain tumor detection"

_PeerJ Computer Science, doi:10.7717/peerj-cs.2670_

## Round 0.1 · original submission · Major Revisions

The article cannot be accepted for publication in its current condition.

The reviewers indicated a series of problems that need to be addressed and solved. Please revise and resubmit

Reviewer 1 ·

Basic reporting

The paper presents a novel deep-learning model for brain tumor detection using MRI data. While the language is generally clear, several areas could be improved to enhance the overall quality of the manuscript.

Introduction and Background: The introduction should clearly state that the study focuses on MRI data. This crucial information should be emphasized both in the abstract and at the beginning of the introduction. Additionally, the numerous subheadings in the introduction could be consolidated to improve readability.

Figures and Tables: The captions for figures and tables are often too brief and do not provide sufficient context. They should be expanded to describe the content of each visual accurately. The font inconsistency in the bibliography should be addressed. Furthermore, the Jupyter Notebook contains errors, and the results presented in the tables should be fully reproducible. The mathematical formulas should be carefully checked for accuracy, paying close attention to subscripts and superscripts.

Experimental design

The research question is well-defined and relevant. However, there are several concerns regarding the experimental design:

1. Dataset augmentation: Two datasets were utilized in this study. The second dataset was subjected to data augmentation techniques to increase its size. Subsequently, all data from both datasets were merged into a single, unified dataset. This combined dataset was then partitioned into training, validation, and testing subsets. However, this methodology poses a potential risk: augmented images originating from the second dataset could inadvertently be included within the test set. As a result, the model might demonstrate superior performance when evaluated on these augmented images, as it would have already encountered similar examples during training. This could lead to an overestimation of the model's generalization ability.

2. Threshold sensitivity: The authors mention that the model's performance depends on the chosen threshold. However, the specific meaning of this threshold and its impact on the results are not clearly explained. Additionally, it is unclear whether the performance of the compared models was evaluated using the same threshold (if applicable).
3. Algorithm structure: It is misleading to include dataset preparation within the algorithm description. Dataset preparation is a pre-processing step and should not be considered part of the algorithm itself.
4. Confidence intervals: The results should include confidence intervals for the reported performance metrics to assess the model's variability accurately.

Validity of the findings

The paper's main contribution lies in proposing a lightweight deep-learning model for brain tumor detection. However, the discussion of the model's computational efficiency could be strengthened. While the authors claim their model is computationally efficient, the evidence supporting this claim is limited.

It is recommended that Tables 2 and 3 be combined to provide a more comprehensive comparison of the proposed model with other state-of-the-art methods. This would allow for a more precise visualization of the trade-offs between accuracy and computational efficiency.

Cite this review as

Reviewer 2 ·

Basic reporting

The authors introduce Lightweight-CancerNet, a new deep learning architecture designed to detect brain tumors efficiently and accurately. The proposed framework utilizes MobileNet architecture as the backbone and NanoDet as the primary detection component. The motivation of the work is to achieve real-time human-computer interaction with efficiency and the ability to detect unknown knowledge. The motivation of the work is to overcome the challenges in achieving real-time human-computer interaction with efficiency and the ability to detect unknown knowledge.

Experimental design

The Experimental design are not rigorous

Validity of the findings

In my opinion, the paper shows some important weaknesses

1.The lack of argumentation regarding the novelty introduced by this work
2. Some inaccuracy in the information provided and in the writing (see Minor comments)
3.The lack of an adequate discussion of the work and its implications in light of the
state of the art and the contribution that their work provides.

Additional comments

Minor comments
- the authors do not introduce or cite ultra-light deep learning frameworks.

- the English is not adequate.

- EVALUATION PARAMETERS >> please check the formula: they are not aligned

- EXPERIMENTAL SETUP AND RESULTS
“Before training, warm-up training is conducted with a step size of 5. Simultaneously, one-dimensional linear interpolation is employed. The cosine annealing algorithm adjusts the learning rate dynamically during the training process.” ….This statement is not clear. The concepts should be introduced. What is “cosine annealing algorithm”? one-dimensional linear interpolation? Please introduce or cite them.

- please check row 178

Cite this review as

---

## Round 0.2 · Minor Revisions

The reviewers raised some new problems regarding this article that the authors need to address. The article cannot be accepted for this publication in its current form. Please address all the indicated problems.

Reviewer 1 ·

Basic reporting

In response to the previous review, the authors have made significant improvements to the manuscript. The paper is now much clearer, more concise, and more rigorous.

The specific improvements, such as the detailed explanation of data preprocessing and the clarification of the model's architecture, have substantially enhanced the paper's quality.

Experimental design

I am particularly pleased with the authors' attention to detail in addressing data augmentation and model evaluation concerns. The revised manuscript provides a more robust and reliable assessment of the model's performance.

Validity of the findings

While the paper has made significant strides, I have a minor comment about the clarity of the threshold's meaning. I encourage the Authors to add a more straightforward explanation of the meaning and why it is so relevant to the problem.

Cite this review as

Reviewer 2 ·

Basic reporting

The quality of paper has been improved. However, I think that the novelty of the paper should be explained better. There are still a lot of typos.
- please check formulas (1) and (2). The formulas should be in a math environment.
- Algorithm 1: Lightweight-CancerNet (Proposed) show some typos.
- EVALUATION PARAMETER. TP, TN, FN, FP in Formulas (6)-(9) should be explained in the text
- in CONCLUSION, This paper presents Lightweight-CancerNet, A new deep-learning framework  This paper presents Lightweight-CancerNet, a new deep-learning framework
In general all formulas shoed be checked.

Experimental design

no comment

Validity of the findings

no comment

Additional comments

no comment

Cite this review as

---

## Round 0.3 · accepted · Accept

The authors correctly addressed the issues raised by the reviewers and therefore I can recommend the article for acceptance and publication.